# Expression of Androgen and Estrogen Receptors in the Human Lacrimal Gland

**DOI:** 10.3390/ijms24065609

**Published:** 2023-03-15

**Authors:** Koraljka Hat, Ana Planinić, Davor Ježek, Snježana Kaštelan

**Affiliations:** 1Department of Maxillofacial Surgery, Clinical Hospital Dubrava, 10000 Zagreb, Croatia; 2Department of Histology and Embryology, School of Medicine, University of Zagreb, 10000 Zagreb, Croatia; 3Scientific Center of Excellence for Reproductive and Regenerative Medicine, 10000 Zagreb, Croatia; 4Department of Ophthalmology, Clinical Hospital Dubrava, School of Medicine, University of Zagreb, 10000 Zagreb, Croatia

**Keywords:** lacrimal gland, sexual dimorphism, sex steroid hormones, androgen receptors, estrogen receptors, dry eye disease, RNA

## Abstract

Lacrimal gland dysfunction causes dry eye disease (DED) due to decreased tear production. Aqueous-deficient DED is more prevalent in women, suggesting that sexual dimorphism of the human lacrimal gland could be a potential cause. Sex steroid hormones are a key factor in the development of sexual dimorphism. This study aimed to quantify estrogen receptor (ER) and androgen receptor (AR) expression in the human lacrimal gland and compare it between sexes. RNA was isolated from 35 human lacrimal gland tissue samples collected from 19 cornea donors. AR, ERα, and ERβ mRNA was identified in all samples, and their expression was quantified using qPCR. Immunohistochemical staining was performed on selected samples to evaluate protein expression of the receptors. ERα mRNA expression was significantly higher than the expression of AR and ERβ. No difference in sex steroid hormone (SSH) receptor mRNA expression was observed between sexes, and no correlation was observed with age. If ERα protein expression is found to be concordant with mRNA expression, it should be investigated further as a potential target for hormone therapy of DED. Further research is needed to elucidate the role of sex steroid hormone receptors in sex-related differences of lacrimal gland structure and disease.

## 1. Introduction

Sexual dimorphism (SD) includes every difference between male and female individuals of the same species not related to reproduction, and is ubiquitous in the living world [1]. Sex steroid hormones (SSHs), which include androgens, estrogens, and progestogens, are a key factor in the development of sexual dimorphism, and their role goes far beyond reproduction, including the regulation of multiple physiological and behavioral functions essential for overall health and longevity [1,2,3,4]. Significant sex-related differences in the anatomy, physiology, and pathophysiology of the lacrimal gland, which can all contribute to variations in the tear film, have been observed in numerous studies [5,6,7,8,9,10,11,12,13,14,15]. Lacrimal glands are paired exocrine glands that represent one of the main components of the tear film-producing lacrimal functional unit (LFU), along with Meibomian glands, accessory lacrimal glands, the cornea, conjunctiva, and interconnecting innervation [16]. The tear film protects the cornea, aids wound healing after injury, and maintains eye comfort and high-quality vision [17]. Disorders affecting any component of the LFU, mainly caused by systemic or local inflammation [16,18], can lead to dry eye disease (DED). DED is a multifactorial disease characterized by tear film instability and/or deficit, causing discomfort and/or visual impairment, accompanied by varying degrees of ocular surface epitheliopathy, inflammation, and neurosensory abnormalities [19]. Two types of DED can be distinguished based on etiopathogenesis: aqueous-deficient dry eye (ATD), characterized by decreased tear production in lacrimal glands, and evaporative dry eye (LADE), characterized by the instability of the tear film caused by Meibomian gland dysfunction [20]. A total of 85.6% of subjects with DED had signs of LADE, or a combination of LADE and ATD [21], suggesting a predominance of Meibomian gland dysfunction in DED [22]. Nevertheless, ATD affects 1.7% of the population and is the only DED subtype more prevalent in women [23]. Since women have about 50–70% higher risk of developing DED [24], with differences even more pronounced after menopause, female sex and increasing age seem to be significant risk factors for DED development [25,26]. As increasing age leads to a decrease in the levels of SSHs, levels of androgens and estrogens could have a significant role in the pathophysiology of lacrimal gland dysfunction. The second most frequent cause of DED is autoimmune diseases, which are significantly more frequent in women. An example is Sjögren’s syndrome, a disorder that primarily affects lacrimal and salivary glands, and over 90% of patients are women [27]. Other autoimmune disorders, such as sarcoidosis and Graves‘ disease, can also cause non-infectious dacryoadenitis and DED.

SSHs, along with genes and the environment, are key factors in the development of sexual dimorphism. However, their functions in the lacrimal gland are still unclear. Optimal androgen levels seem to be essential for the normal function of lacrimal and Meibomian glands [28,29,30]. Previous studies of estrogen effects on the lacrimal gland have failed to report uniform results, some suggesting that estrogens have a pro-inflammatory effect, others that the effect is anti-inflammatory, while some did not find any effect of estrogens on the lacrimal gland [24]. It appears that estrogens and progestogens, compared to androgens, contribute relatively little to sex-related differences in gene expression and sexual dimorphism of the lacrimal gland. However, further investigation is warranted to elucidate their roles [31,32]. Steroidogenic enzyme mRNAs have also been detected in the lacrimal gland, which could implicate that it contains enzymes for intracrine synthesis and metabolism of sex steroids. This could explain why the relationship between estrogens and DED appears to be unclear [33]. Androgen deficiency has been demonstrated in women with Sjögren’s syndrome, and it appears to facilitate disease progression [34]. Estrogen seems to reduce inflammation in mouse models of Sjögren’s syndrome, but contradictory results were observed in a study that found that hormone replacement therapy in post-menopausal women increased the risk of dry eye syndrome [35,36]. This could be due to the opposite effect of high and low doses of estrogen on inflammation. In low doses, estrogen seems to promote gland cell survival and protect against exocrine gland inflammation, while high levels of estrogen can increase inflammation [37]. Lacrimal gland deficiency has been observed in women after menopause or ovariectomy and in women taking oral contraceptives, despite variable estrogen levels [38,39]. Men taking anti-androgen medications, on the other hand, do not show changes in tear secretion, which suggests that androgen action on the lacrimal gland could be sex-specific [40]. Hormone replacement therapy (HRT) with estrogen and medroxyprogesterone acetate appears to be effective in increasing tear production, but not necessarily tear quality [41,42]. Testosterone and dihydrotestosterone appear to be effective in rats, but there has not been sufficient evidence to warrant a trial in humans [43].

Steroid hormone receptors are transcription factors belonging to the nuclear receptor subfamily three, which includes a group of estrogen receptors and a group of 3-ketosteroid receptors (GR, MR, PR, AR). Testosterone and dihydrotestosterone are ligands for AR, while estrogens (estrone-E1, 17β-estradiol-E2, and estriol-E3) act through ER (ERα, ERβ, and GPER). SSHs act predominantly as intranuclear ligand-activated transcription factors, leading to slow transcription-dependent genomic actions [44,45,46,47]. The integration of all steroid actions in the cell coordinates outcomes such as cell fate, proliferation, differentiation, and migration [48,49,50,51,52,53]. AR is primarily expressed in reproductive organs, while ERα is primarily found in female genitalia, but both can also be detected in various tissues. ERβ is present in low levels in various tissues [54,55,56,57,58]. ERα and β are products of different genes, and although they share a high degree of amino acid sequence homology and bind to the same estrogen response element (ERE) on DNA [59], they exhibit different tissue distribution and have different biological effects [49,60]. The regulation of sex steroid receptor expression is complex and is affected by various factors, including age and sex. AR, ERα, and ERβ have been detected in the human lacrimal gland by immunohistochemistry [26,61]. AR, ERα, and ERβ mRNA was also detected, but was never quantified or analyzed for sexual dimorphism [29,61,62]. Sexual dimorphism was determined only for AR expression in rat lacrimal glands [63]. The objective of this study was to fill that gap and quantify mRNA expression of AR, ERα, and ERβ in the human lacrimal gland, compare the levels of expression in males and females, and correlate them with the patient’s age.

## 2. Results

### 2.1. Sex Steroid Receptor mRNA Expression in the Human Lacrimal Gland

AR, ERα, and ERβ mRNA was detected in all tissue samples. Relative ERα expression was found to be significantly higher than that of AR and ERβ. There was no significant difference in the expression of AR and ERβ. The same was observed in both sexes (Figure 1). AR, ERα, and ERβ expression was not significantly different between sexes. No significant difference in ERα to ERβ, ERα to AR, or AR to ERβ ratios was observed between sexes. The ratios were calculated by dividing the relative expression of the receptors in question. (Figure 1).

### 2.2. Correlation between Sex Steroid Receptor mRNA Expression in the Human Lacrimal Gland

The expression of AR and ERβ showed a moderate positive correlation (*p* < 0.001), while ERα did not show a significant correlation with the expression of other sex steroid receptors (Figure 2).

### 2.3. Impact of Age on Sex Steroid Receptor mRNA Expression in the Human Lacrimal Gland

No significant correlation was observed between AR, ERα, or ERβ expression with age (Figure 3). No correlation was observed between ERα to AR, AR to ERβ, or ERα to ERβ ratios and age. A strong correlation was observed between the ERα to ERβ and ERα to AR ratios, which was expected, due to the already observed correlation between AR and ERβ expression (Figure 4).

### 2.4. Expression of AR and ERα Proteins in the Human Lacrimal Gland

To evaluate if the mRNA we detected in the human lacrimal gland is translated into proteins, we performed immunohistochemical (IHC) staining on four selected samples (two male and two female) in which mRNA expression was already determined. AR and ERα signals were observed in all samples. Both AR and ERα proteins are generally observed in the nucleus, which is where we detected them in our control samples (cervix for ERα and prostate for AR). In the lacrimal gland, a faint AR signal was observed in some nuclei of acinar cells, while a much stronger ERα signal was observed in the cytoplasm, as well as the nuclei, of acinar cells (Figure 5 and Figure 6). The signal intensity was quantified for both AR and ERα, and ERα signal intensity was significantly higher than that of AR. This result is concordant with the analysis of mRNA expression in the same samples (Figure 7).

## 3. Discussion

Our study has confirmed mRNA expression of AR, ERα and ERβ in the human lacrimal glands. Sex steroid receptor expression in the human lacrimal gland was quantified for the first time, and ERα mRNA expression was significantly higher than mRNA expression of AR and ERβ. No difference was found in sex steroid receptor expression between sexes, and no correlation was observed with age. Protein expression observed in preliminary IHC experiments seems to correlate with mRNA expression.

Significantly higher relative mRNA expression of ERα compared to AR and ERβ in both male and female samples suggests a predominance of ERα in the human lacrimal gland. Significant differences in ERα and ERβ mRNA expression have already been documented in numerous human [55,64] and animal tissues [65]. ERα and β are products of different genes and exhibit tissue and cell-type-specific expression. They are co-expressed in several tissues and share the same general structure, including a ligand-binding domain, a DNA-binding domain, and two activation function (AF) domains [66]. Although they share a high degree of amino acid sequence homology and bind to the same estrogen response element (ERE) on DNA [59], ERα and β exhibit different tissue distribution and different biological effects [49,60]. The biological roles of the mutual existence of ERα and β are still unknown [67]. It is believed that ERβ exhibits an inhibitory action on ERα-mediated gene expression and, in many instances, opposes the actions of ERα in the case of their co-expression [55,68]. The results of our preliminary IHC protein expression analysis seem to be concordant with the mRNA expression analysis, but they should be confirmed on a larger sample. The ERα signal was observed in the cytoplasm as well as the nucleus, which differs from the usual signal pattern of ERα, though a similar pattern was observed in a previous IHC study of ERα expression in the lacrimal gland [61]. The fact that ERα mRNA is the dominantly expressed subtype of ER mRNA in the human lacrimal gland could be important for the future design of targeted selective hormonal therapy, if confirmed with a more extensive protein expression analysis, preferably with Western blot [55].

Our results did not show any significant differences in mRNA expression of AR, ERα, and ERβ between sexes. The absence of sex differences in SSH receptor expression implicates the greater importance of other factors in the development of sexual dimorphism and the epidemiology of DED [69]. It has been suggested that most sex differences are consequences of variations in gene expression [70], and SSHs are important modulators of gene expression [71,72]. In the absence of sex hormones, sex chromosomes proved to be insufficient in inducing sexual dimorphism in lacrimal glands [73]. So, it seems that SSH receptors are involved in the development of sexual dimorphism in the human lacrimal gland, but many factors affect their level of expression. Studies on ovariectomized and orchidectomised rats have shown equal expression of androgen receptors in both sexes, while the intact male rats had a significantly higher number of androgen receptors compared to female rats, demonstrating that androgens can autoregulate their own binding sites [28,29]. Moreover, the excess or deficiency of steroid hormones regulates the number of active receptors through up- and downregulation of the expression of their receptors, as well as the receptors of other sex hormones [48]. Levels of serum testosterone in men physiologically start decreasing slowly after the age of 30, at a rate of about 1% per year [74]; however, a more significant decrease is only seen in older age [75]. Testosterone levels in women also begin to decline after the age of 30 and can decline further during menopause [30,76]. Estrogen levels fluctuate during the menstrual cycle, with the highest levels before ovulation, and they start to decline significantly during menopause, which typically occurs between the ages of 45 and 55. SSH serum levels also fluctuate diurnally, seasonally, in pregnancy, and under the influence of psychological factors, all of which could affect the expression of SSH receptors [77]. Since the lacrimal gland also contains enzymes for steroid synthesis, intracrine signaling could be an important factor in regulating lacrimal gland function, and the reason why the relationship between SSH serum concentrations and SSH receptor expression is not straightforward. It seems that due to the complex regulation of SSH receptor expression, there is no observable difference in expression between sexes. The results also imply that there are factors other than SSH receptor expression that cause the higher prevalence of DED in older women. While lower levels of testosterone and estrogen during perimenopause could be a factor, it does not have to be reflected in SSH receptor expression due to complex regulation. Other factors, such as the use of makeup and skincare products, contact lenses, as well as environmental factors such as air conditioning, heating, and pollution, could impact the prevalence of DED in women and should be investigated further.

No significant sex-related difference was observed in the ratios of SSH receptor expression. The AR/ER ratio has been shown to be associated with outcomes in ER+ breast cancer patients [78]. The ERα/ERβ ratio within a cell may determine the cell’s sensitivity to estrogens and its biological response to the hormone [55]. Comparing ratios can normalize sample-to-sample variation, reduce noise and biases in the study, and better illustrate the biology of the signaling pathways if they are influenced by relative levels of proteins [79].

AR, ERα, and ERβ expression did not show a significant correlation with age. This could be due to the fact that the median age of the donors in our study was 70 years and the younger population was not represented adequately. Lower SSH levels in the older age group could lead to SSH receptor downregulation and, consequently, loss of potential age differences in SSH receptor expression. Degenerative changes could also compromise adequate tissue collection due to a heterogenous histological structure, with some samples containing less lacrimal tissue and more connective tissue. These findings again implicate that the expression of SSH receptors in the lacrimal gland is regulated by multiple factors.

A moderate positive correlation was observed between AR and ERβ expression, while a strong positive correlation was observed between ERα to ERβ, and ERα to AR ratios. We analyzed the relationship between AR and ERβ expression in all human tissues using the Correlation AnalyzeR application [80], and the Pearson’s coefficient was 0.125, implicating that AR and ERβ expression is generally not correlated. The Pearson’s coefficient for AR and ERβ expression calculated from our data was 0.63, suggesting that some of the genes’ common pathways could be active in the human lacrimal gland.

Our study is the first that quantified SSH receptor mRNA expression in the human lacrimal gland, a tissue sample that is rare and difficult to obtain. The main limitation of our study is that we could not perform eye exams and relevant diagnostic tests on patients, since we collected the samples post-mortem. We checked the full patient history for all patients, and none had a history of DED or were ever treated for it, but this does not exclude undiagnosed DED which could, to an extent, mask sex-related differences that exist between healthy individuals. Further studies of AR and ER expression in patients with and without DED are needed to elucidate the roles of SSH receptors in the pathophysiology of this disease. An additional limitation of our study is donor age. Most tissue samples originated from older donors with a median donor age of 70 (27–89 years), and all samples were obtained post-mortem. Future research should include participants of reproductive age so that the effect of menopausal and andropausal changes in hormone levels can be examined. Another limitation is that we determined the relative amount of mRNA in the tissue and, although protein and mRNA levels typically show a reasonable correlation, it is unclear if the transcription of a certain gene leads to an increased concentration of the target protein [64]. Our preliminary IHC analysis shows that there could be a correlation between mRNA and protein expression for AR and ERα, but further investigation is needed.

Although sex differences are mainly consequences of variations in gene expression [70], and SSHs are important modulators of gene expression, our study has failed to prove any sexual dimorphism in SSH receptor mRNA expression, or any impact of age on their expression. An unexpected finding was the predominance of ERα mRNA in both male and female human lacrimal gland samples, which could point to ERα as a potentially important target for future research of selective hormonal therapy. Further research on mRNA expression of SSH receptors on more participants of a wider age range is needed to elucidate the role of the receptors in the pathogenesis of human lacrimal gland disorders. Protein expression analysis, ideally via Western blot, should be conducted to confirm that this difference in mRNA expression is also observed at the protein level and could, thus, be clinically relevant.

## 4. Materials and Methods

### 4.1. Sample Collection

Lacrimal gland samples were collected from 19 cornea donors (*n* = 19, 10 female and 9 male), who underwent the cornea harvesting procedure for the Eye Bank of the University Hospital Zagreb, Croatia. The median donor age was 70 (27–89 years). The median donor age was also 70 for female donors (*n* = 10, 36–89 years) and for male donors (*n* = 9, 27–82 years). Inclusion criteria for donors were: explantation within 24 h from the moment of death and age over 18 years. Exclusion criteria were patient history of DED, acute infection in the orbital area, systemic infection (sepsis, HIV, CMV, hepatitis C, COVID-19), Sjögren’s syndrome, IgG4-related disease, inflammatory orbital pseudotumor, chronic GVHD, sarcoidosis, insulin-dependent diabetes, previous surgery in the orbital area, radiation in the head and neck area, hormone replacement treatment, treatment with systemic corticosteroid, and hormone therapy for oncological patients. Both lacrimal glands were explanted from each donor. Explantation of the glands was performed using the transconjunctival approach. Tissue samples for planned molecular studies were harvested from both explanted glands immediately after the procedure and frozen at −80 °C. The remaining tissue underwent fixation in 10% neutral buffered formalin, dehydration, and was embedded in paraffin. Explantation of both lacrimal glands was performed in the case of 16 donors, with one sample from each gland being analyzed. In the case of three donors, only one sample was available.

### 4.2. RNA Extraction, Reverse Transcription, and Real-Time PCR

Total RNA was extracted using the RNeasy Mini Kit (Qiagen, Hilden, Germany). High-Capacity cDNA Reverse Transcription Kit (Applied Biosystems, Waltham, MA, USA) was used for reverse transcription. The real-time PCR approach used the SYBR green method in a 96-well plate on a CFX96 Touch Deep Well Real-Time PCR Detection System. Reactions contained 5 uL PowerUp™ SYBR™ Green Master Mix (Applied Biosystems, Waltham, MA, USA), primer (100 nM), and template in a total volume of 10 µL. Melting curve analysis was performed on the products of the amplification phase. The thermocycler program was: 95 °C for 3 min (denaturation) followed by 40 cycles. Each cycle consisted of 95 °C for 10 s and 60 °C for 30 s.

The primers were:5′-GCCTTGCTCTCTAGCCTCAA-3′ (f) and5′-GGTCGTCCACGTTAAGTTG-3′ (r) for AR;5′- CCAGGGAAGCTACTGTTTGC -3′ (f) and5′-TGATGTAGCCAGCAGCATGT -3′ (r) for ERα;5′-GCTGAACGCCGTGACCGATGCT-3′ (f) and5′-CCCGTGATGGAGGACTTGC-3′ (r) for ERβ;5′TCAACGACCACTTTGTCAAGC-3′ (f) and5′GGTGGTCCAGGGGTC-3′ (r) for GAPHD.

Negative controls containing no cDNA were subjected to the same procedure as tissue samples to exclude contamination. The absolute value of each gene’s mRNA expression was normalized to that of GAPHD, the housekeeping control gene using the 2^−∆Ct^ method.

### 4.3. Immunohistochemistry (IHC)

Slides with lacrimal gland sections from 4 patients (2 male and 2 female) were selected and incubated in a thermostat at 55 °C, and then were further deparaffinized in a xylol solution. The sections were then rehydrated in a descending series of alcohol (100%, 96%, and 70% EtOH), and the samples were then washed in distilled H_2_O. Heat-induced epitope retrieval was performed using the Tris-EDTA Buffer with a pH of 9.0 in a steamer for 60 min, followed by 30 min of cooling at room temperature (RT). The slides were then washed in 1 × PBS buffer. Non-specific binding was blocked with 2.5% normal horse serum for 20 min. The primary antibodies were diluted in PBS buffer containing 0.1% Triton X-100 and 1% normal horse serum. The Anti-Estrogen Receptor alpha antibody (ab3575, Abcam, Cambridge, UK) was used in a 1:250 dilution, while the Anti-Androgen Receptor antibody (ab133273, Abcam, Cambridge, UK) was used in a 1:100 dilution. Primary antibody incubation was performed in a humid chamber at 4 °C overnight. The samples were then washed in PBS buffer and the Goat anti-rabbit IgG Amplifier antibody (DK-1594, Vector Laboratories, Burlingame, CA, USA) was applied for 15 min. After washing the slides in PBS, the VectaFluor DyLight labeled Horse Anti-Goat IgG (DK-1594, Vector Laboratories, Burlingame, CA, USA) secondary antibody was added. After washing in PBS, tissue sections were treated with TrueBlack^®^ lipofuscin autofluorescence quencher solution (Biotium, Fremont, CA, USA), which was diluted 20× with 70% EtOH, to reduce the background fluorescence. Samples were also stained with a Hoechst solution and then covered with a mounting medium (Vectashield, Vector Laboratories, Burlingame, CA, USA). Confocal microscopy images were acquired using an Olympus FV1000 microscope with FV10-ASW software, Version 4.2b. All the acquisition parameters were always kept the same. The figures were processed using ImageJ. Signal intensity was quantified on ROIs determined using the moments algorithm. The mean grey value was determined, and the background intensity determined on negative controls was subtracted.

### 4.4. Statistics

An analysis of the normality of the distribution of numerical data was performed using the Kolmogorov–Smirnov test. Corresponding parametric or non-parametric statistical analyses and methods of data presentation were applied to the obtained results. AR expression followed a normal distribution, so the independent samples *t*-test was used. ERα and ERβ expression differed significantly from a normal distribution, so the non-parametric Mann–Whitney U test was used. Spearman’s non-parametric correlation coefficient for age distribution was calculated. All *p*-values under 0.05 were considered statistically significant.

## Figures and Tables

**Figure 1 ijms-24-05609-f001:**
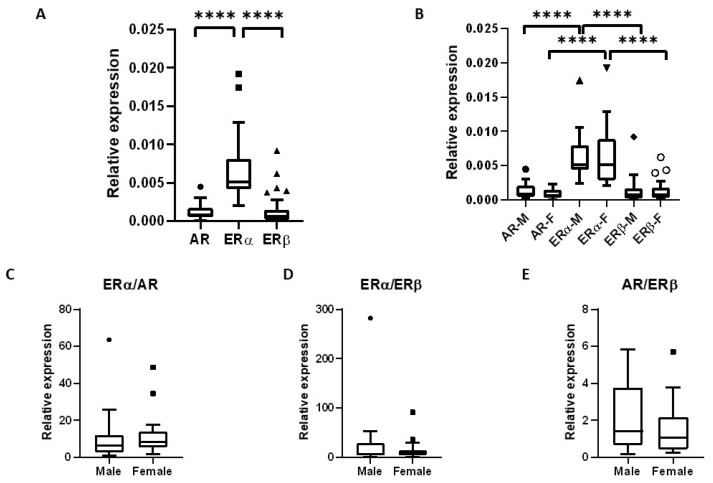
Sex steroid receptor mRNA expression in human lacrimal glands. (**A**) Box and whiskers plot that shows significantly higher ERα expression in comparison to AR and ERβ expression, and no significant difference between AR and ERβ expression in lacrimal gland samples. (**B**) Box and whiskers plot that shows no significant difference between male and female lacrimal glands in AR, ERα and ERβ expression. ERα expression is significantly higher than AR and ERβ expression in both male (M) and female (F) samples. (**C**–**E**) Box and whiskers plots that show no significant difference in ERα to AR, ERα to ERβ, and AR to ERβ expression ratios, respectively, between male and female lacrimal glands. **** *p* < 0.0001.

**Figure 2 ijms-24-05609-f002:**
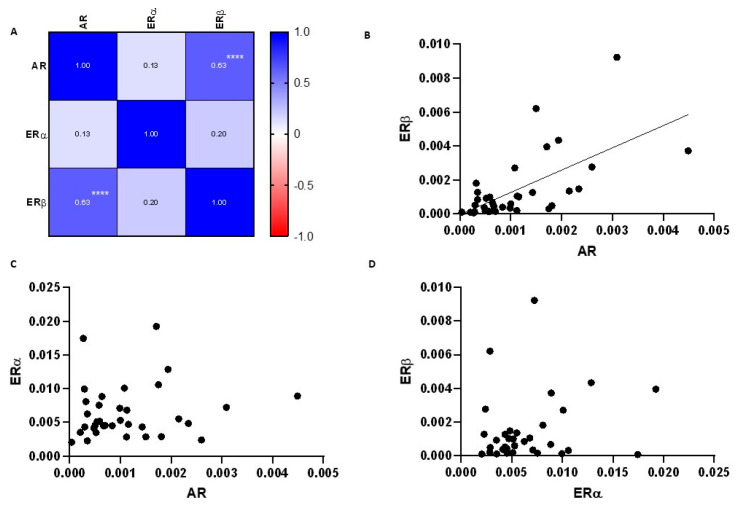
Correlation between steroid receptor mRNA expression in the lacrimal gland. (**A**) Correlation matrix showing correlation coefficients between sex steroid receptor expression. (**B**) Scatter plot with simple linear regression showing a moderate positive correlation between ERβ and AR expression. (**C**,**D**) Scatter plots showing the relationship between ERα and AR, and ERα and ERβ expression, respectively. **** *p* < 0.0001.

**Figure 3 ijms-24-05609-f003:**
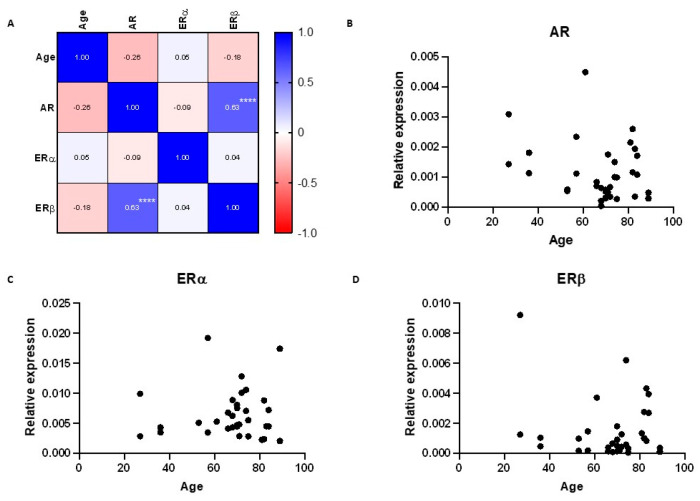
Correlation between the mRNA expression of sex steroid receptors in the human lacrimal gland and correlation of their expression with age. (**A**) Correlation matrix showing correlation coefficients between age and sex steroid receptor expression. (**B**–**D**) Scatter plots showing the relationships between patients’ age and AR, ERα, and ERβ expression, respectively. **** *p* < 0.0001.

**Figure 4 ijms-24-05609-f004:**
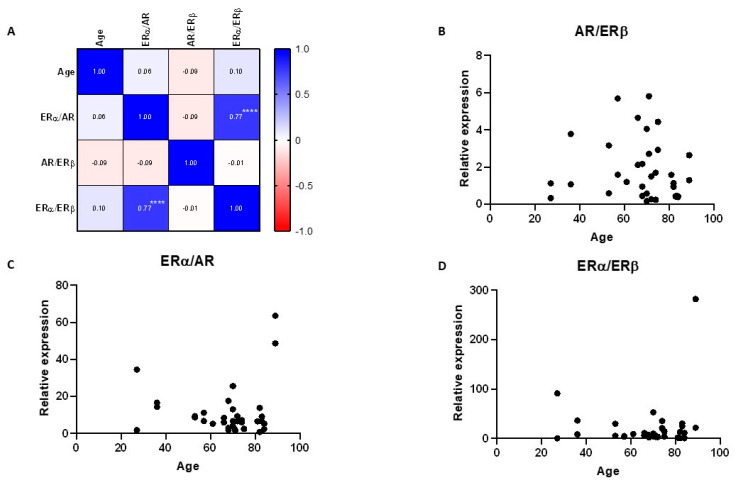
Correlation of sex steroid receptor mRNA expression ratios in human lacrimal glands with age. (**A**) Correlation matrix showing correlation coefficients between age and sex steroid receptor expression ratios; (**B**–**D**) Scatter plots showing the relationships between age and AR to ERβ, ERα to AR, or ERα to ERβ ratios. **** *p* < 0.0001.

**Figure 5 ijms-24-05609-f005:**
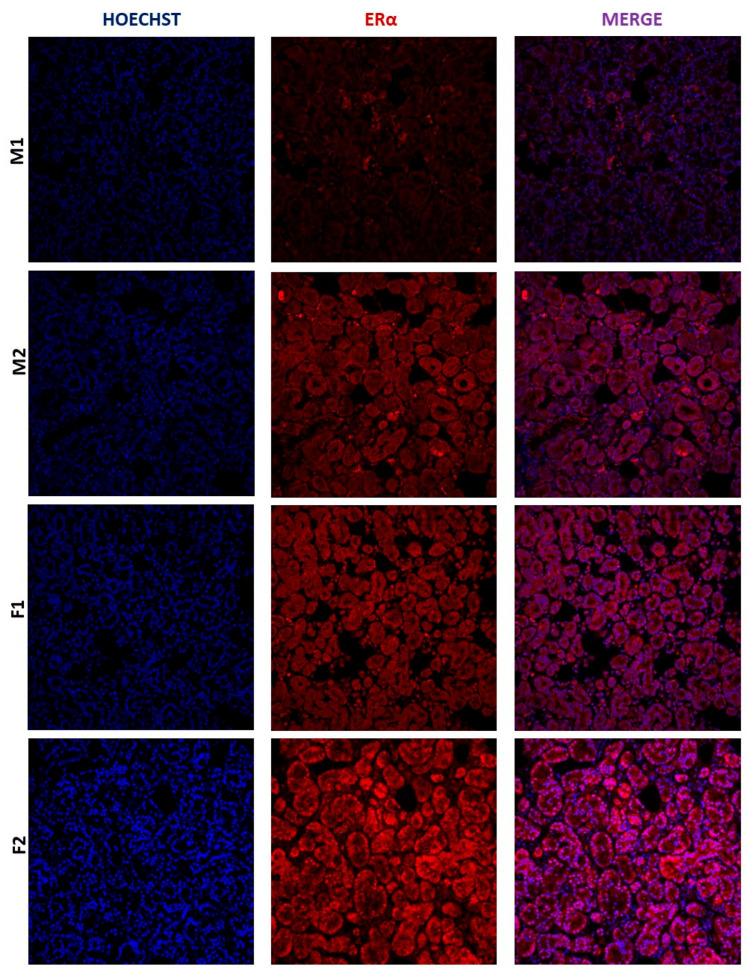
ERα expression in the human lacrimal gland, magnification 20×. (M1—male lacrimal gland sample 1, M2-male lacrimal gland sample 2, F1—female lacrimal gland sample 1, F2—female lacrimal gland sample 2).

**Figure 6 ijms-24-05609-f006:**
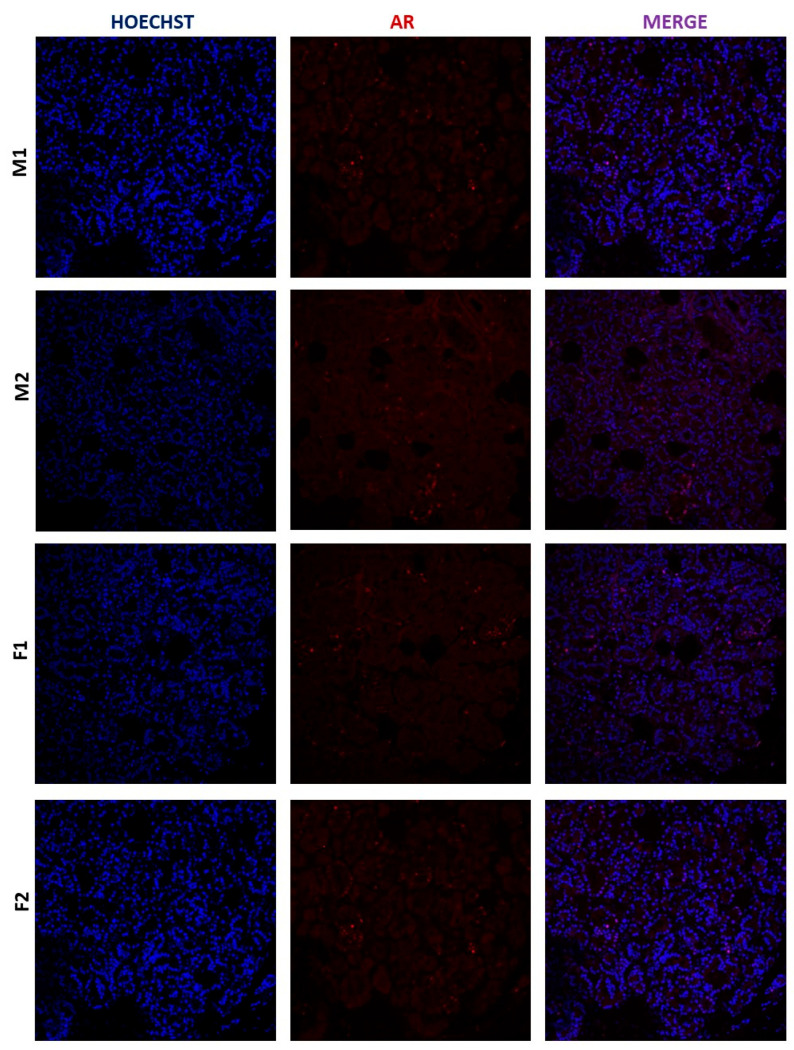
AR expression in the human lacrimal gland, magnification 20×. (M1—male lacrimal gland sample 1, M2-male lacrimal gland sample 2, F1—female lacrimal gland sample 1, F2- female lacrimal gland sample 2).

**Figure 7 ijms-24-05609-f007:**
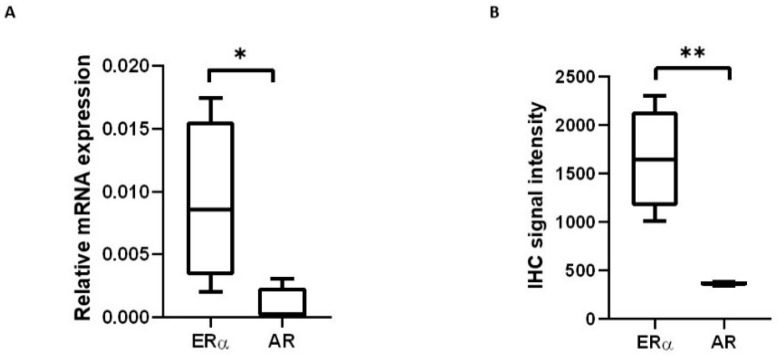
Comparison of mRNA and protein expression of ERα and AR in selected human lacrimal gland samples. (**A**) Box plot showing significantly higher mRNA expression of ERα than AR in lacrimal gland samples. (**B**) Box plot showing significantly higher ERα than AR IHC signal intensity in lacrimal gland samples. * *p* < 0.05, ** *p* < 0.01.

## Data Availability

Not applicable.

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
