# Peer review of "Expression of Androgen and Estrogen Receptors in the Human Lacrimal Gland"

_ijms, 2023, doi:10.3390/ijms24065609_

Round 1

Reviewer 1 Report (Previous Reviewer 2)

It is a highly original and interesting research. Thank you very much for letting me review it. With the addition of IHC results, we can understand the new development of the sex hormone receptor in the lacrimal gland. It would be nice to delve a little deeper into that and the epidemiology of dryeye in the discussion.

1

For example, there are many characteristics in women, especially postmenopausal women, and consideration of this finding

2 How does estrogen supplementation affect acinar destruction and lymphocyte infiltration in autoimmune diseases such as Sjögren's syndrome?

Author Response

Reviewer 2 Report (New Reviewer)

The authors did a great job exploring the expression of androgen and estrogen receptors in the human lacrimal gland. However, I have a few comments that might improve the manuscript

1) The introduction and discussion are lengthy and should be shortened. 

2) Figure 4 needs to be mentioned in the text

3)  Why ER beta was not investigated by IHC too?

4) The repetition of some information made the manuscript too long. For example, " The ratio of ER α vs. β within a cell may determine the cell’s sensitivity to estrogens and its biological response to the hormone" sentence was repeated several times.

5) The paragraphs in the discussion section need to adjust. For example, the first paragraph on page 9 was discussing the effect of sexual dimorphism then age then dimorphism again on SSH. It is confusing and each paragraph needs a specific topic.

6) what is the minimum age in your study? was it male or female? this point was not written correctly in the material and methods. please revise.

Author Response

Reviewer 3 Report (New Reviewer)

The authors of Manuscript ID: ijms-2252513, presented interesting research entitled  „Expression of androgen and estrogen receptors in the human  lacrimal gland” by Hat et al. This study aimed to quantify estrogen receptor (ER) and androgen receptor (AR) expression in the human lacrimal gland and compare it between sexes. RNA was isolated from 35 human lacrimal gland tissue samples collected from 19 cornea donors. AR, ERα and ERβ mRNA was identified in all samples and their expression was quantified using qPCR. The authors showed that ERα mRNA expression was significantly higher than AR and ERβ expression, but no sex hormone receptor (SHH) mRNA expression differences or correlation with age were observed. This is an important study because the role of sex steroid hormone receptors sex-related differences of lacrimal gland structure needs to be clarified. Introduction - the authors have prepared the research topic exhaustively Material and methods: two methods. RNA extraction was performed reverse transcription and real-time PCR and Immunohistochemistry. Results - prepared very reliably. They are presented in the form of figures as well as colorful immunohistochemistry. The authors also conducted a thorough discussion supported by well-chosen literature. In my opinion this is a very well prepared manuscript and should be accepted for publication in its current form.    

Author Response

This manuscript is a resubmission of an earlier submission. The following is a list of the peer review reports and author responses from that submission.

Round 1

Reviewer 1 Report

In this manuscript the authors investigate the expression levels of sex hormones and their main receptors in the human lacrimal gland in order to find a scientific justification to why aqueous deficient DED is more prevalent in older women. Although this is a necessary and highly relevant study, there is insufficient data to be able to draw conclusions. This study does have enormous merit in having been able to obtain human LGs for analyzing the expression levels, but unfortunately real time PCR alone is not enough for a study of this nature. Ideally additional samples needed to be obtained to corroborate the findings at the protein level, for example immunostaining or western blotting. The entire paper is based on a small set of real time PCR data for 5 genes. The authors provide a number of figures based on the same data with various types on statistical analysis and correlations.

Major issues:

1.     The authors make the statement that the expression of ERalpha is significantly higher than that of AR and ERbeta. Based on what data to the authors make this claim? The affinity of primers for their products affects the number of copies that are amplified, so it is not possible to make comparisons between the expression levels of different genes purely based on real time PCR data. Ideally, western blotting analysis would be carried out to back up this statement. If the authors would like to leave this statement they should change it to clearly state how this conclusion was drawn and indicate that it is a speculation.

2.     Since the same data is shown in Figure 1 and 2, the authors should consider consolidating them into the same figure.

Minor issues:

1.     The manuscript does require English revision, particularly in the introduction.

Reviewer 2 Report

This is a very interesting initiative, but since there are no clinical parameters and only mRNA is being studied,

If only 1mRNA is used, the evidence as knowledge is weak, so I think it would be good to perform immunostaining and show the distribution of ER.

2. I have the impression that dry eye is more common in elderly women, but why was there no significant difference between age and gender?

Round 2

Reviewer 1 Report

The manuscript Hat et al., analyzes the expression levels of sex hormone receptors in human LGs in order to investigate potential sex relates differences that could have a role in dry eye diseases.

Major issues

1.     The authors must include the immunohistochemistry data in the manuscript.

2.     In Figure 1 the authors do not make it clear how the values in c-e were calculated and what the usefulness of this data is.

3.     The authors have not found any significant differences in AR or ER receptor expression that could explain the increased prevalence of DED in females. The main deficiency in this study would be that the samples have not been separated based on whether the individual had DED or not. The fact that DED patients and controls have been grouped in this study could be masking any sex related differences that could exists between healthy and/or diseased males and females.

Minor

1.     Include a sentence in the introduction to explain the differences between Era and ERb.

2.     There are still some grammatical errors throughout the text.

Reviewer 2 Report

Very well corrected.

This article is suitable for publication.
